# 'Care co-ordinator in my pocket': a feasibility study of mobile assessment and therapy for psychosis (TechCare)

Nadeem Gire [1,2] Neil Caton,[1] Mick McKeown [3] Naeem Mohmed,[4] Joy Duxbury,[5] James Kelly,[6] Miv Riley,[1] Peter J Taylor,[7] Christopher D J Taylor,[8] Farooq Naeem,[9] Imran Bashir Chaudhry,[10,11] Nusrat Husain[1,12]

For numbered affiliations see end of article.

**Correspondence to**
Dr Nadeem Gire;
ngire1@uclan.ac.uk

## ABSTRACT

**Objectives** The aim of the project was to examine the acceptability and feasibility of a mobile phone application-based intervention 'TechCare', for individuals with psychosis in the North West of England. The main objectives were to determine whether appropriate individuals could be identified and recruited to the study and whether the TechCare App would be an acceptable intervention for individuals with psychosis.

**Methods** This was a mixed methods feasibility study, consisting of a test-run and feasibility evaluation of the TechCare App intervention.

**Setting** Early Intervention Services (EIS) for psychosis, within an NHS Trust in the North West of England.

**Participants** Sixteen participants (test-run n=4, feasibility study n=12) aged between 18 and 65 years recruited from the East, Central and North Lancashire EIS.

**Intervention** A 6-week intervention, with the TechCare App assessing participants' symptoms and responses in real-time and providing a personalised-guided self-help-based psychological intervention based on the principles of Cognitive Behaviorual Therapy (CBT).

**Results** A total of 83.33% (n=10) of participants completed the 6-week feasibility study, with 70% of completers achieving the set compliance threshold of ≥33% engagement with the TechCare App system. Analysis of the qualitative data suggested that participants held the view that the TechCare was both an acceptable and feasible means of delivering interventions in real-time.

**Conclusion** Innovative digital clinical technologies, such as the TechCare App, have the potential to increase access to psychological interventions, reduce health inequality and promote self-management with a real-time intervention, through enabling access to mental health resources in a stigma-free, evidence-based and time-independent manner.

**Trial registration number** ClinicalTrials.gov Identifier: NCT02439619.

## BACKGROUND

The use of mobile devices (eg, mobile phones) for the delivery of healthcare interventions, referred to as mHealth, is a growing field globally with the potential to improve mental health. Research in the area of mHealth has indicated the scope

### Strengths and limitations of this study

► This study is the first to examine the intelligent Real Time Therapy conceptual model for psychosis.
► Participants met 83% of all follow-up data points, indicating a good retention rate for the study.
► Service user engagement was an integral part of the research design, with service users consulting on the design and development of the App.
► The study was carried out in only one Early Intervention Service for psychosis in the Northwest of England.
► The availability of the intervention in additional languages may have allowed for the inclusion of a more representative sample.

to develop mobile phone interventions, which look at the assessment and treatment of psychiatric disorders in real-time.[1–4] The use of mobile devices could provide greater autonomy to service users[5]; some of whom otherwise may be seen as a 'hard to engage group', with complex relationships between psychotic experiences, trust and engagement with services. mHealth may be able to offer a non-stigmatising approach to treatment through providing a discrete medium for seeking support, which can be both accessible and anonymous, as mental health stigma is one of the biggest barriers to engagement with mental health services,[6–8] which can compound difficulties with treatment adherence and, thus, outcomes.

Early Intervention Services (EIS) were introduced into the England National Health Service (NHS) in the 1990s for people with a first episode of psychosis.[9–12] The primary aim of EIS was to reduce the delay in the duration of untreated psychosis, with the rationale being that early treatment of psychosis could result in a greater chance of recovery and a reduction in the adverse psychosocial impact

of the illness.[13 14] Previous research has shown that these services are cost-effective and are successful in reducing relapse and reductions in hospital admissions.[15 16]

Digital technologies, which include self-help strategies promoting service user autonomy and control, may potentially assist in making the resources of EIS go further, supplementing face-to-face practitioner time with service users. This is important due to the increasing levels of pressure on mental health services as a result of substantial reductions in resources dedicated to mental health services.[17] There are waiting lists to receive psychological therapies in many EIS, mobile technology may be an alternate way of engaging and supporting service users, by enhancing self-management and increasing the accessibility of psychological support. The TechCare App was, thus, developed to provide real-time therapy[18 19] targeted at reducing symptoms of psychosis, allowing appropriate interventions to occur in real-time and thereby reducing the possibility of relapse. Improving the access to digital clinical technologies, particularly in low resourced or deprived localities can be of great benefit in providing a step-change in the utilisation of low-cost mobile technologies.[20]

This study looked to integrate a momentary sampling assessment approach that is matched with a momentary basic psychological intervention to address low mood and paranoia. This feasibility work intended to inform the design of a future larger Randomised Controlled Trial (RCT) and examine important parameters such as the identification of appropriate outcome measures, follow-up periods and estimates of recruitment and feasibility of the TechCare App intervention.

## Aim

The aim of the project was to develop and conduct a feasibility study of the mobile phone Application (App) 'TechCare' for individuals with psychosis in the North West of England.

The specific objectives of the research were to:
1. To determine whether a small sample of eligible (n=16) individuals could be identified and recruited to a study for the evaluation of TechCare for psychosis.
2. Whether TechCare would be an acceptable intervention for individuals with psychosis? To be determined by qualitative interviews.
3. Whether service users can be engaged in setting goals and reporting outcomes to their care coordinators and work towards the goals using the TechCare App. This will be determined by the extent of engagement with the app.
4. Establish the most appropriate primary outcome measure for a future randomised RCT and cost-effectiveness trial of the TechCare intervention. To be determined by the ease of use of assessment measures and participant preference.

## METHODS
### Study design
The feasibility study followed the National Institute of Health Research[21] guidance on feasibility study design and used a mixed methods approach. The feasibility study included a test-run and qualitative semistructured interviews. The TechCare App assessed participants' symptoms and responses and provided a personalised-guided self-help-based psychological intervention, with the aim of reducing participants' symptoms and enhancing their coping abilities. An initial, test-run with a small number of participating service users (n=4) was conducted, to refine the mobile App intervention (TechCare) and address any technical issues found in the App software, content and design, based on participants' feedback. The TechCare App was then evaluated as part of a feasibility study with a total of n=12 service users. In addition, a total of 16 pre-intervention qualitative interviews and 13 post-intervention interviews were completed. The study was preregistered on ClinicalTrials.gov and ethical approval was obtained from the National Research Ethics Service (NRES) Committee North West—Preston REC reference: 14/NW/1192.

### Participants
The sample for the study was recruited from the East, Central and North Lancashire EIS for psychosis teams in the North West of England, between August 2016 and October 2017. Potential participants were volunteers who had already shown interest in the study, plus additional service users, carers and EIS care coordinators were invited to take part in the study. The participants, who took part in the test-run and feasibility study, were individuals under the care of the EIS. The inclusion and exclusion criteria for the study were as follows:

### Inclusion criteria
► Each service user must have been accepted into the Psychosis Group of the Lancashire EIS.
► Age 18–65 years.
► Medication stable for the previous 2 months; the EIS uses a traffic light system to indicate current symptomatology and risks of each client. For this study, we only included service users with a Green Light, signifying that their mental state was stable.
► A score of 3 or more on positive symptoms on the Positive and Negative Syndrome Scale (PANSS)[22]
► Minimum score of 1 on the Calgary Depression Scale (CDS)[23]

### Exclusion criteria
► Patients with drug-induced psychosis, an acquired brain injury or moderate to severe learning disability, as determined by the service users treating clinician.
► Service users who were undergoing assessment, not formally diagnosed and accepted into the service.
► Lacking capacity for informed consent.

► At Risk Mental State group (ie, prodromal, not first episode).

## Defining the intervention: experiential sampling methodology and intelligent real-time therapy

The TechCare App software was developed specifically for use on a smartphone device, requiring a touchscreen interface. The mobile phone App would alert participants via notifications and ask a series of questions. Based on the participants' responses, the App would provide a tailored Cognitive Behavioural Therapy (CBT)-based intervention, which could include participants' preferred multimedia such as music, images or videos. The App used Experiential Sampling Methodology (ESM) as a research methodology, which allowed participants to record, subjective experiences in real-time of their thoughts, moods and experiences of distress. The ESM research methodology was coupled with intelligent Real-Time Therapy (iRTT), which is a conceptual model developed by Kelly et al,[18] which uses the data gathered by ESM on a participants' subjective experiences of distress and in response provides interventions to be delivered in real-time. The iRTT may comprise of several formats, including media, images or MP3 formats which were all integrated in the present study.

Three notifications were sent between the time points, 10:00 and 22:00 each day over the study period. If the App detected low mood/paranoia, participants were offered tailored interventions based on the iRTT model. The system would then renotify the participants a total of three times, every 60 min if low mood/paranoia was detected over a period of ≃4 hours. If symptoms persisted for greater than 4 hours, this would initiate a prior agreed response being displayed on the App. Crisis planning is a routine part of EIS treatment, where service users would work collaboratively with their care coordinators to agree a plan of action to follow when in crisis. When symptoms are exacerbated causing severe distress, the crisis response may include contacting the EIS or an agreed designated contact. In the feasibility context, an examination of the response rates to the questions and notifications was observed including the participants' selection of interventions.

The TechCare App ESM and iRTT system used intelligence at two levels. The first level involved recognising and intelligently increasing the frequency of assessment notifications if low mood/paranoia was detected. This was done via feedback loops monitoring symptoms over time and the deployment of a tailored crisis plan if there was a prolonged period of low mood/paranoia detected (≃4 hour). The second level included an intelligent machine learning algorithm, providing interventions in real-time when assessment thresholds were exceeded (eg, when levels of paranoia exceeded a certain threshold), thus providing recommendations on the most popular interventions selected by the cohort of participants on the study, listed in rank order. See Husain et al[19] for further details of the methodology.

## Outcome measures

The primary aim of the study was to determine the feasibility and acceptability of the TechCare App intervention. This included measuring whether eligible individuals could be identified and recruited to a feasibility study of the TechCare App. The success criterion for feasibility was the recruitment of ≥50% of eligible participants, based on the recruitment rate of previous research using the ClinTouch system,[5] which this study was based on. In addition, the acceptability of the intervention was assessed based on the amount of engagement and usage of the TechCare App, with the success criterion for compliance being set at ≥33%. In the ClinTouch study[5] compliance was calculated as engagement with 33% of the available notifications. In addition, the following outcome measures were used; PANSS,[22] Psychotic Symptom Rating Scales (PSYRATS),[24] satisfaction with CBT therapy on the CHoice of Outcome In CBT for psychosis,[25] Mental well-being on the Warwick-Edinburgh Mental Well Being Scale,[26] Measure of core beliefs regarding self and others on the Brief Core Schema Scale,[27] Depression on the CDS,[23] Work and social functioning on The Work and Social Adjustment Scale[28] and Quality of life on the EuroQoL-5 Dimensions[29] to determine the most appropriate outcome measures for a future clinical and cost-effectiveness trial.

## Patient and public involvement

The project was developed with patient and public involvement planned from the outset, with key input from service users who consulted on the development and refinement of the App. Our service user representatives (NC and NM) were involved in the design and layout of the TechCare App, with NC attending the research ethics committee meeting. NC and NM were actively involved in the design of the study, interpretation of study findings and contributed to the development of the manuscript.

As this was a mixed methods feasibility study, both quantitative and qualitative data were gathered and analysed. The quantitative data collected as part of the study are presented using summary statistics (SD, mean, CIs). Preliminary analysis was performed to compare the baseline and post-intervention scores on the outcome measures. In the feasibility context, data on recruitment and retention were calculated, with the iRTT data gathered allowing for the analysis of participant responses and the selection of interventions during the study. Semistructured interviews were conducted using a topic guide to collect the qualitative data. The semistructured interviews were transcribed verbatim and Framework Analysis was used to analyse the data.[30] The Framework Analysis started with the process of familiarisation, where data from all transcripts were read a number of times to gain an understanding and familiarity with the content. The next stage of the analysis involved key ideas and themes that were recurring being noted, with the final themes being compiled into a thematic framework. The Framework Analysis was conducted to look for emerging

themes, focusing on feasibility, acceptability and further development of the intervention.

## RESULTS

### Test-run results

Feedback from participants in the test-run provided insight into the refinement and further development of the App, both in terms of the intervention content, design, research process and procedures. A total of four participants took part in the test-run, with three participants being men and one woman. The mean age of the participants was 22.5 years old (SD=4.39). In addition, we examined any software difficulties or faults that occurred during the test run. The software code was monitored by the software engineers, with errors in the code being sourced and corrected. Changes in the software were mainly in relation to ensuring the notification system that was working correctly, and the screen size and login system were appropriately configured.

### Feasibility study results

A total of n=28 participants were approached to take part in the study with n=16 consenting to take part. However, two participants dropped out; one participant was unable to continue with the study at week 2 due to not being concordant with medication, and one participant decided to drop out at week 2 due to not wanting to proceed. No explanation was given as to the reasons behind the latter drop out. The recruitment rate was calculated at 57% of the total number of participants approached in the study. Overall, 83% of those who consented to take part in the feasibility study completed the study. Figure 1 shows the consort flow diagram, outlining the recruitment and retention rates of both the test run and feasibility study.

The feasibility study sample comprised of 12 participants; all participants were aged between 25 and 35 years. The mean age of the participants was 24.83 year (SD=4.83). The sample consisted of 8 men and 4 women, 9 were unemployed, 7 were single, with all 12 having access to the internet. All but one had access to a smartphone, with the participant being loaned a mobile phone for the duration of the study (see table 1).

### Acceptability: engagement and usage of the TechCare App intervention

The TechCare system notified the participants three times per day (a minimum of 126 notifications over the 6-week period). Compliance was based on engagement with the App, at least, a total of 42 times over the study period (≥33%). It was found that out of 12 participants, eight (66.67%) achieved compliance; however, this figure takes into account the two participants who dropped out after week 1. On removing these two participants, out of the 10 remaining participants, a total of 70% achieved compliance, related to answering the TechCare App questions.

In addition, the intervention screen was shown a total of 82 times across the 6-week period with the most selected

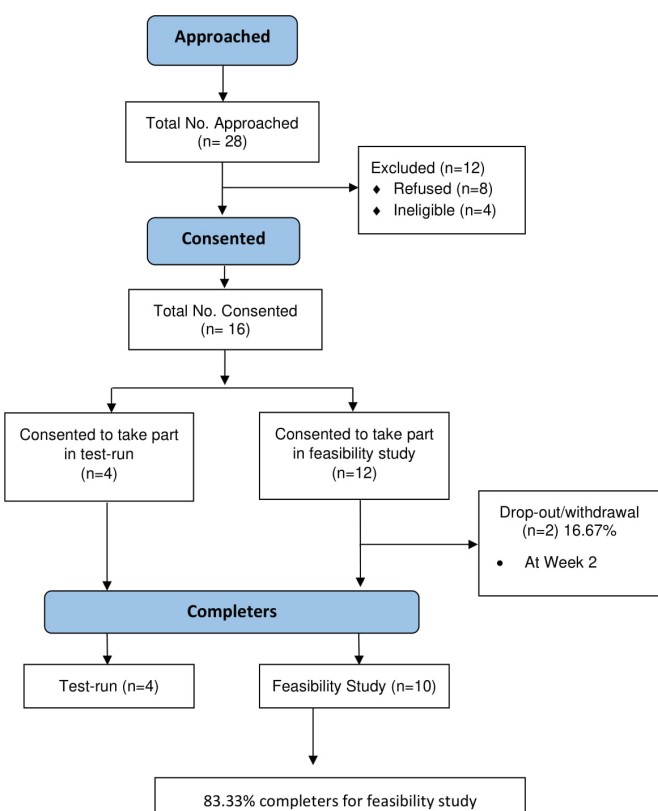

**Figure 1** Consort flow diagram to show recruitment and retention.

intervention being the multimedia intervention. Data collected from the online server provided insight into the day-to-day usage of the App by each of the participants. There were a number of key variables that were analysed to provide a descriptive account of the App usage across the feasibility study period. Overall, we found that out of the 12 participants, the App was registered and loaded a total of 947 times with participants using the App on average 1.88 times per day.

The TechCare system also allowed for the collection of patient derived response data through the TechCare App. The data were scored on a 1–7 Likert scale, with one being completely disagree and seven being in full agreement with the TechCare App questions related to symptoms of low mood and paranoia (see Husain *et al*[19]). The average weekly score in week 1 for the Depression scale was M=29.13 (SD=18.29) and for week 6 was M=17.50 (SD=11.92), which indicates a decrease in depressive symptoms from week 1 to week 6. Furthermore, there was a similar trend on the paranoia scale, with the average score decreasing from week 1 (M=38.00, SD=28.27) to week 6 (M=33.92, SD=27.88). On analysis of the notification system data, it was found that over the 6-week period, participants clicked on the notifications a total of 521 times, with the average number of questions completed by participants being 5.63 times per day (range: 0–25). Furthermore, it was found that the participants used the self-help material a total of 114 times. This was the psychoeducational information tab located on the home

**Table 1** Demographic for participants in the feasibility study

| Feasibility study | Total |
|---|---|
| Gender | |
| Male | 8 |
| Female | 4 |
| Age | |
| Mean age | 24.83 |
| Range | 19–35 |
| Ethnicity | |
| British—White | 8 |
| British—Indian | 2 |
| British—Pakistani | 2 |
| Work | |
| Self-employed | 0 |
| Part-time employment | 1 |
| Unemployed | 9 |
| Student | 2 |
| Living situation | |
| Living with family | 10 |
| Lives on own | 1 |
| Lives in shared accommodation | 1 |
| Marital status | |
| Single | 7 |
| Partner/married | 4 |
| Separated/divorced | 1 |

screen. These findings suggest engagement with the App and the feasibility of the system. One interesting finding from the results was that none of the participants reached the threshold for the crisis intervention.

## Outcome assessment results

The mean scores on both the PANSS and PSYRATS were calculated at baseline ((PANSS Positive Scale (M=18.33, SD=3.81; 95% CI 16.41 to 21.25), PANSS Negative Scale (M=18.00, SD=7.45; 95% CI 13.27 to 22.73), PANSS General Psychopathology (M=34.58, SD=4.91; 95% CI 31.47 to 37.70), PSYRATS Voices (M=12.75, SD=12.48; 95% CI 4.82 to 20.68) and PSYRATS Delusions (M=9.17, SD=11.43; 95% CI 12.76 to 16.91) and at week 6 (end of intervention) (PANSS Positive Scale (M=12.50, SD=7.06; 95% CI 8.01 to 16.99), PANSS Negative Scale (M=1167, SD=7.97; 95% CI 6.60 to 16.73), PANSS General Psychopathology (M=22.75, SD=12.85; 95% CI 14.59 to 30.91), PSYRATS Voices (M=14.83, SD=3.27; 95% CI 1.90 to 16.43) and PSYRATS Delusions (M=3.75, SD=7.11; 95% CI −0.77 to 8.27)). In addition to the App preintervention and post-intervention measures, face-to-face weekly assessments were conducted with the participants (see table 2). The weekly outcome measures were collected over the 6-week intervention period. It can be seen that there was

a reduction in mean scores over the 6-week period on the CDS. However, the other weekly measures examined suggested that there was a decrease in scores until week 5, with week 6 scores increasing slightly. Furthermore, the assessment scores seem to decrease from weeks 1–3 then plateau during weeks 4–6. This may have been due to increased interest and engagement with the App during the beginning of the intervention period (weeks 1–3) and this diminishing over the latter half of the intervention period (weeks 4–6).

## Qualitative study results
### Pre-intervention qualitative one-to-one interviews with service users

The qualitative interviews investigated the feasibility and potential acceptability of the intervention. Overall, the key themes that were identified on the analysis of the pre-intervention qualitative interviews (n=16), were organised into a coding framework as follows: accessing support for psychosis, mobile phone usage and ownership, the acceptability of the TechCare intervention, confidentiality and security and finally areas of development and refinement of the App.

### (1) Accessing support

This theme presents participants' views regarding accessing support for their mental health difficulties, specifically highlighting difficulties in accessing support. The theme explored the individuals' experiences of various factors that impede access to appropriate and valued support. These included an overall limited access to services, a lack of understanding of their needs, the role of stigma in accessing services and experience of feelings of isolation. Throughout, the participants focused attention to possible benefits that a helpful mobile App may provide in addressing the above difficulties.

P10: I found in the past that … to be able to get the right help that you need is very difficult because you have to end up going through so many different people. It sometimes wears you down until you get to the right person

P4: I had names called because of illness … a phone is discrete no one will know

P8: It's a good idea yeah, because like you said there's a lot of stigma around it, with mental health issue, like people start looking at you differently

### (2) Mobile phone usage

This theme portrayed the feeling that although mobile devices were an important part of day-to-day life, there was a need for moderation and some face-to-face contact. Mobile devices were reported to be easily accessible, with all participants engaging in mobile phone usage throughout the day. This inferred that mobile devices could potentially be a familiar medium for accessing support. Furthermore, the participants

**Table 2** Mean and SD for the weekly assessment measures

| | | Calgary Depression Scale | Brief Core Schema Scale (negative self) | Brief Core Schema Scale (positive self) | Brief Core Schema Scale (negative self) | Brief Core Schema Scale (positive other) | Work and Social Adjustment Scale | Warwick-Edinburgh Mental Wellbeing Scale | Choices | EQ5-D |
|---|---|---|---|---|---|---|---|---|---|---|
| Week1 (baseline) | Mean | 9.06 | 8.00 | 5.17 | 9.42 | 9.42 | 20.44 | 37.94 | 48.06 | 5.75 |
| | SD | 5.51 | 5.58 | 4.58 | 7.34 | 5.59 | 10.23 | 10.39 | 30.02 | 2.64 |
| Week 2 | Mean | 5.75 | 5.40 | 7.50 | 8.50 | 7.50 | 17.94 | 33.69 | 60.56 | 10.88 |
| | SD | 5.80 | 5.32 | 5.09 | 7.88 | 5.77 | 11.50 | 16.04 | 58.62 | 23.62 |
| Week 3 | Mean | 6.06 | 7.00 | 8.56 | 9.22 | 10.00 | 17.25 | 33.69 | 55.19 | 9.66 |
| | SD | 5.80 | 7.91 | 4.95 | 8.59 | 6.12 | 12.16 | 20.11 | 45.66 | 24.00 |
| Week 4 | Mean | 2.63 | 5.44 | 7.44 | 6.56 | 12.33 | 12.25 | 21.81 | 31.00 | 3.26 |
| | SD | 4.43 | 6.56 | 5.09 | 7.55 | 7.28 | 13.85 | 21.30 | 33.33 | 3.29 |
| Week 5 | Mean | 2.13 | 2.88 | 11.88 | 5.25 | 12.71 | 8.69 | 23.88 | 34.00 | 3.73 |
| | SD | 3.50 | 4.05 | 3.40 | 6.85 | 5.70 | 11.66 | 25.54 | 38.90 | 3.87 |
| Week 6 | Mean | 2.13 | 5.80 | 10.00 | 7.30 | 11.70 | 11.13 | 27.75 | 39.88 | 4.08 |
| | SD | 3.32 | 7.31 | 6.31 | 7.68 | 6.90 | 13.38 | 24.57 | 40.45 | 3.64 |
| Total | Mean | 4.63 | 5.97 | 7.78 | 7.96 | 9.77 | 14.61 | 29.79 | 44.78 | 6.22 |
| | SD | 5.37 | 5.57 | 4.70 | 7.25 | 5.58 | 12.59 | 20.62 | 42.48 | 13.97 |

EQ5-D, EuroQoL–5 Dimensions.

also talked about the financial implications of mobile ownership and connectivity to the internet.

> P16: Texting is easier, sort of still need face-to-face as it helps to build a dialogue and is empowering

In addition, social media was found to be a useful aspect of the experience of going online. However, there were some concerns that you could post things online and you could be judged by people.

### (3) Acceptability of the TechCare App intervention

The App was described as a 'brilliant idea' by participants, and that treatments delivered in this way would be acceptable. All participants provided feedback suggesting that they did not envisage service users in general being averse to using the TechCare App. However, factors such as the App layout and design features were crucial areas for consideration.

> P3: [TechCare Logo] it's like anonymous, you can't tell … You can't tell what it is, if it was like a medicine sign people would know what it means

### (4) Confidentiality and Security

Participant views relating to confidentiality were also discussed in the one-to-one interviews. There was specific reference to the security access on the device and how it can have an impact. In addition, it was recommended that having a privacy policy on the App would provide further details on the confidentiality arrangements of the intervention. Having this explained by a health professional would help reassure service users about the confidentiality of data.

> P9: Umm because I know some people like me who might not want our information to be [shown] to everybody … um I think if it was explained by a care worker or… yeah umm maybe have the privacy policy on the App
>
> P12: Confidentiality, you should put like a password on or something

### (5) Areas for development and refinement of the App

There were a number of ideas proposed by the participants in relation to the further development and refinement of the App. Some of the areas suggested by the service users were enabling personalisable settings within the TechCare App, inclusion of helpful websites, suicide prevention support helplines, calendar reminders for medication and appointments with health professionals, the ability for service users to note down how they are feeling using the App and information on mental illness and medication side effects.

> P15: For medication reminders maybe, I need like a video of the service, so you know what to expect
>
> P8: Yeah I suppose. Sometimes I struggle like it's when, like a week or whatever between my appointments, its

hard work remembering things that have gone on, so I suppose it keeps you up to date, it keeps it live

In addition, there was a general consensus that service users had a limited understanding of psychosis and had difficulties in explaining their experiences to family and friends. Recommendations were made to include helpful information on the App, relating to gaining a better understanding of psychosis.

> P4: People's experiences/stories what other people are going through
>
> P9: Things like videos explaining mental illness
>
> P8: When I got diagnosed with psychotic disorder, I looked it up online and there are loads of psychotic disorders, so I was thinking which ones me … it's just all very broad and vague

Other development and refinement ideas presented by the participants centred around the inclusion of coping strategies and motivational pictures.

> P16: Coping strategy information about medication side-effects
>
> P9: Yeah I think there is a positive side to it, for example like motivational pictures. Things like pictures, like that have motivational quotes on them

The findings from the pre-intervention qualitative work informed the further development and refinement the TechCare App in preparation for the feasibility study.

### Post-intervention qualitative interviews with service users

On completion of the test-run and feasibility study, follow-up one-to-one interviews (n=13) were conducted. Participants provided feedback on the overall acceptability of taking part in the TechCare research study, with the experience being enjoyable and empowering.

### (1) Acceptability and feasibility

The participants provided their views relating to their experience of using the App and the consensus was that it was an acceptable method for receiving psychological interventions. The views on research related procedures included length of time taken to complete assessments and the research recruitment procedures, which were considered acceptable.

> P12: Overall, I think the App was a really good idea, it's a new way of doing treatments and it works alongside your medication
>
> P15: The process of the using the TechCare App was empowering and was an achievement as I normally struggle to come outside
>
> P12: Overall, I think the App was a really good idea, it's a new way of doing treatments and it works alongside your medication

## (2) Usability and user experience

The usability of the device in terms of its day-to-day usage was found to be easy to manage with particular reference to the easy navigation of the TechCare App. This was suggested to be an important factor in the App usage. It was also highlighted that the psychoeducational links were a useful tool to understand specific information on psychosis.

> P12: It was very easy to use, very easy, very simple, there wasn't any obstacles using it or anything, think it was made very simple, which is a good thing

> P9: Yeah like the time taken to complete the questions … I liked how quick it was

## (3) Accessing and engaging with support

The participants' view was that having easy access to information and useful contact details provided an avenue for participants to seek help. The availability of help through the App was seen as a prompt for service users. The participants reported that the App was used at times when they experienced distress associated with their illness, thus providing greater autonomy and more choice. Talking about mobile devices was suggested to be an important means of starting conversations and providing an alternative way of communicating with health professionals. Although the App was seen as an important tool in accessing support in real-time, the participants were of the view that face-to-face contact was an integral part of the care they received from the EIS service.

> P11: Understanding the thoughts, helps start conversations and stuff just for the fact that I ended up with a new phone just to use for the time being, just started conversations

> P10: Having a face-to-face interview can explore different things like why you are actually feeling down and stuff

## (4) Suggestions for improving the TechCare App intervention

The participants offered a number of suggestions to improve the TechCare App. The improvements related mainly to the App content, such as incorporating a calendar and in-App progress tracking. It was highlighted that mobile technology was part of the future, in providing support for individuals with mental health difficulties, but it was not the whole picture. The areas of improvement included novel ideas such as inclusion of a newsfeed where people with similar experiences could comment on strategies they had used and benefitted from.

> P4: Newsfeed so other people can respond to it like, when I'm feeling down I do this

> P15: The functionality, asking questions tracking the feedback … have reminders early in the day

## (5) Insights into the iRTT system

A key area of focus was testing the iRTT concept. In the view of participants, a number of interventions were used.

Some of the more popular interventions used were multimedia, problem-solving and the use of links to mental health support websites and psychoeducation. The participants held a consensus view that the iRTT system had allowed them to gain insight into their experiences and allowed them to manage their symptoms.

> P12: Before I was depending a lot on [Care coordinator] all the time, whenever something went wrong, but since I got the app, it's been like I have a care coordinator in my pocket, so it felt like there was a mental health professional in my pocket. So whenever I have like a problem I just go on the screen and it would give me solutions which I don't think about at the time … now I don't rely on the App as much, because I've kind of programmed it in my mind, how to, like if I'm facing difficulties how to step-by-step break the problem down

> P13: Helped organise my thoughts … helped me to understand … what I was going through

Overall, it is important to note that participants considered the TechCare App intervention to be acceptable and feasible intervention. The research procedures and processes were also acceptable.

## DISCUSSION

The study provided insights into the development of mHealth interventions for psychosis, which used the iRTT conceptual model. The main objective of the study was related to determining the recruitment, retention, participant dropout rates and engagement with the TechCare App intervention. The results of the study met the success criterion for feasibility and acceptability of the TechCare study. Overall, the key finding of the study was the acceptability and feasibility of the use of the TechCare App and study design, as well as the methods for evaluation. The availability of the intervention in real-time, in contrast to limited face-to-face time with the therapist and the flexibility in the use of the intervention, could be seen as a potential advantage. However, a significant proportion of the participants held the view that this should be in addition to face-to-face contact, rather than a replacement.

Support for these findings comes from a study by Lester et al,[9] showing the importance of technology in engagement within EIS. This is important in drawing emphasis on how mHealth functions could provide continued support and give service users an opportunity for self-management. Indeed, future research may determine the value of mHealth for different groups of service users, distinguishing between service users who engage or not engage with services. It may be that mobile technologies offer different forms of value to each group, with some responding similarly to participants in this study, while some may prefer using technologies to maximise independence and minimum contact with services. Both scenarios raise questions about the nature and experience

of self-management in the context of services operating within the current austerity climate of the UK. The pressure on services is likely to be more in the post-pandemic period.

In addition, another important finding of the feasibility study was that none of the participants reached the threshold for the crisis intervention, which may have been due to inclusion of participants who were deemed clinically stable by their care team. Towards the end of the intervention there was an increase in symptoms and relatively less use of the App. This may be due to the questions that were asked, not being sensitive enough to pick up issues such as boredom with the App or lack of perceived benefit over the 6-week period. It is important to note that the data collected in real-time provided an insight into the day-to-day experiences of the participants, with TechCare providing a flexible and tailor-made self-help intervention, if low mood or paranoia was detected. Participants were able to use this data retrospectively to recollect events in the past week, providing the service user participants greater insight into their experience of their illness.

Current services often expect service users to retrospectively recollect experiences over the preceding week or further back. However, often due to the distress experienced by service users, difficulties in recollection of experiences can occur and difficulties in communicating distress have been found to be a confounding factor when undertaking therapeutic work in psychosis.[5 31] The TechCare App provided an alternative means of recording the participant's experiences and allowed health professionals to view any changes in the symptomatology of the service users, thus overcoming difficulties in recollection of experiences. The results of the study support previous research,[32–34] with ESM being used as an important means of gaining understanding of the real-world socioenvironmental factors related to psychotic symptoms. The participants gave examples of how the App had assisted in their personal recovery journey. However, it should be noted that the project involved weekly assessments, which offers an opportunity for greater levels of engagement with participants.

### Strengths and limitations of the research

The main strength of the study was the feasibility testing of a novel intervention for psychosis, which used the iRTT conceptual model.[18] To our knowledge, iRTT has not been evaluated in an early intervention setting before and as such, the study provided novel insights into the development of mHealth interventions. A further strength of the study was the engagement of the service users, across the study period, with only two participants dropping out. In addition, as we chose a pragmatic approach to the research, which was conducted within the NHS context, this allowed the TechCare App to be feasibility tested as a pragmatic solution to increasing access to psychological therapies. There was good retention with participants meeting 83% of all follow-up data points. Furthermore,

the study only addressed low mood/paranoia in this assessment of feasibility, future research will examine the full spectrum of symptoms of psychosis such as hallucinations and negative symptoms.

One of the limitations of the study was a small sample size (n=16). Although the results of the qualitative work were promising, we present these with caution given the multiple factors associated with routine treatment. We found no adverse events experienced by the participants in the study. This could have been due to the safety considerations made in conjunction with participants' case managers when making referrals to the study and the continued involvement during the study indicating a potential selection bias. However, it is important to note that due to the developing nature of the mHealth field and relatively limited evidence base on the side effects of mHealth Apps,[35] the adverse events may not have been apparent. The study was conducted in only one EIS; therefore, the results cannot be generalised to other EIS in the country.

### CONCLUSIONS

Innovative digital clinical, technologies such as the Tech-Care App, may have the potential to increase service access, reduce health inequality and promote self-management with a real-time intervention. The mobile intervention may also support medication adherence and appointment attendance, in addition to recognising early warning signs both by the participants themselves and also their EIS care coordinators. The concept of self-management and self-help is interesting in that they appear to chime in with a valuing of independence and individual autonomy that fits well with a recovery ethos within healthcare services. Service users emphasised a balance between the positive aspects of self-reliance, a more collective, network-based psychosocial support system and a continued value of face-to-face therapeutic relations with skilled health professionals. In addition, mHealth can play a major role in low-resourced settings, especially in areas where there are limited funds and resources to spend on healthcare. Measuring trends and analysing real-time real-world data may also allow for better forecasting and ensure measures can be put in place to improve clinical practice and more efficient use of limited resources. Following this feasibility study, relevant alterations will be made to the TechCare App, and based on this feasibility and acceptability data, we plan to submit a funding application for a larger appropriately powered, trial with an internal pilot to investigate the clinical and cost-effectiveness of the TechCare App intervention

**Author affiliations**
[1]Research, Lancashire Care NHS Trust, Preston, UK
[2]School of Medicine, University of Central Lancashire, Preston, UK
[3]School of Nursing, University of Central Lancashire, Preston, UK
[4]Research and Development, Lancashire Care NHS Foundation Trust, Blackburn, UK

[5]Faculty of Health, Psychology & Social Care, Manchester Metropolitan University, Manchester, UK
[6]Faculty of Health and Medicine, Lancaster University, Lancaster, UK
[7]Psychology and Mental Health, University of Manchester School of Psychological Sciences, Manchester, UK
[8]Secondary Care Psychological Therapies Service, Pennine Care NHS Foundation Trust, Ashton-under-Lyne, UK
[9]Psychiatry, University of Toronto Faculty of Medicine, Toronto, Ontario, Canada
[10]Psychiatry, Ziauddin University, Karachi, Pakistan
[11]Faculty of Medicine, Biology and Health, University of Manchester, Manchester, UK
[12]School of Health Sciences, Division of Psychology & Mental Health, The University of Manchester, Manchester, UK

**Acknowledgements** NG gratefully acknowledges the support and funding of his PhD Studentship by the NIHR Collaboration for Leadership in Applied Health Research and Care (CLAHRC), North West Coast, UK and Lancashire and South Cumbria NHS Foundation Trust. We would also like to acknowledge the input and support of Andrew Holland at Maywoods Ltd, in the development and design of the TechCare App.

**Contributors** NG, NH, IBC, JK conceived the idea. MM, JD, NC, NM, CDJT contributed to the design of the project. NC, NM, PJT, FN helped develop the research tools. NG led data collection and oversaw the recruitment process. NG, MM led the analysis of the qualitative data and quantitative data. NG, MM, NH, IBC drafted the initial manuscript. All authors read and approved the final manuscript. NH is the guarantor.

**Funding** This report is independent research funded by the National Institute for Health Research Applied Research Collaboration North West Coast (ARC NWC) and Lancashire and South Cumbria NHS Foundation Trust (grant number: NIHR-CLAHRC-NWC-007). The views expressed in this publication are those of the authors and not necessarily those of the National Institute for Health Research or the Department of Health and Social Care.

**Competing interests** IBC and NH have given lectures and advice to Eli Lilly, Bristol Myers Squibb, Lundbeck, Astra Zeneca and Janssen Pharmaceuticals, for which they or their employing institution have been reimbursed. IBC and NH were previously trustees of the Pakistan Institute of Learning and Living. NH is Chair of the board of trustees of the Manchester Global Foundation. CDJT reports grants from UK National Institute for Health Research (NIHR) Research Fellowship award (DRF-2012-05-211), delivers psychological therapy in the UK National Health Service and personal fees from providing occasional workshops on CBT, outside the submitted work. The other authors have no competing interests to declare.

**Patient and public involvement** Patients and/or the public were involved in the design, or conduct, or reporting, or dissemination plans of this research. Refer to the Methods section for further details.

**Patient consent for publication** Consent obtained directly from patient(s)

**Ethics approval** Ethical Approval was obtained from the NRES Committee North West - Preston REC reference: 14/NW/1192.

**Provenance and peer review** Not commissioned; externally peer reviewed.

**Data availability statement** Data are available upon reasonable request. The anonymised data is available upon reasonable request.

**ORCID iDs**
Nadeem Gire http://orcid.org/0000-0002-4130-6626
Mick McKeown http://orcid.org/0000-0003-0235-1923

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
