## [Reviewer comments · BMJ Open]

ARTICLE DETAILS

TITLE (PROVISIONAL)	'Care Co-ordinator In My Pocket'. A Feasibility study of Mobile-Assessment and Therapy for Psychosis (TechCare).
AUTHORS	Gire, Nadeem ; Caton, Neil; McKeown, Mick; Mohmed, Naeem; Duxbury, Joy; Kelly, James; Riley, Miv; Taylor, Peter; Taylor, Christopher; Naeem, Farooq; Chaudhry, Imran; Husain, Nusrat

VERSION 1 – REVIEW

REVIEWER	Kingdon, David University of Southampton, Department of Psychiatry I was previously aware of this study and have worked with participants in the past. I was not however in anyway involved in the conduct of it.
REVIEW RETURNED	26-Mar-2021

GENERAL COMMENTS	Reference Originality: this is a highly original study which looks at providing app-based therapy for individuals in early intervention teams. There are promising studies in existence of relapse prevention that this goes beyond them. Importance: provision of treatment early in the course of psychosis is necessary to reduce distress and disability and has been demonstrated to improve outcomes but psychological treatment in particular is rarely or insufficiently available to appropriate standards. It is therefore very important that a choice of effective methods of delivery are developed which can be used across all settings internationally. It is now apparent that mobile phone technology is becoming available in most societies and is very acceptable to younger people to whom these interventions are particularly relevant. Technology can also allow continuing support after individual therapy to sustain gains which is also emerging as an important issue for EIP. Methodology: assessment of feasibility is an essential first step to evaluating new technologies and this commences with uptake and frequency of usage. Therefore the approach taken is appropriate to the assessment of an emerging technology. Numbers in the study are small but this is appropriate to the type of study proposed.
---

Findings: usage at 84% is good and continuation across the six weeks is reasonable. PANSS and PSYRATs were collected but not reported: there needs to be a comment as to why this is not the case. Details of completion of questionnaires is relevant to a feasibility study. Depression scores are available and it may be that a more specific measure for psychotic symptoms e.g. PSYRATs delusions scale, is more appropriate than broad psychosis measures in any future study.

Conclusions: the study succeeds in its aim and it is appropriate to now consider seeking funding for a larger study.

Limitations: The intervention does not appear to address auditory hallucinations or negative symptoms: this may be perfectly appropriate but needs to be specified and, for the future, inclusion/exclusion criteria adjusted.

'A randomised controlled trial design with a larger sample size, may have strengthened the research design, providing data on the acceptability of the allocation and randomisation procedures' - this is not really a limitation as it wouldn't have strengthened the design at this stage of development.

VERSION 1 – AUTHOR RESPONSE

Reviewer 1:

1) PANSS and PSYRATs were collected but not reported: there needs to be a comment as to why this is not the case. Details of completion of questionnaires is relevant to a feasibility study. Depression scores are available and it may be that a more specific measure for psychotic symptoms e.g. PSYRATs delusions scale, is more appropriate than broad psychosis measures in any future study.

Response: We have now amended the results section to include the pre- and post PANSS and PSYRATS scores. The PANSS and PSYRATS scores are reported with the mean, standard deviation, and confidence intervals.

Amendment: Outcome Assessment Results: The mean scores on both the PANSS and PSYRATS was calculated at baseline ((PANSS Positive Scale (M=18.33, SD=3.81; 95% CI, 16.41 to 21.25), PANSS Negative Scale (M=18.00, SD= 7.45; 95% CI, 13.27 to 22.73), PANSS General Psychopathology (M=34.58, SD=4.91; 95% CI, 31.47 to 37.70), PSYRATS Voices (M=12.75, SD= 12.48; 95% CI, 4.82 to 20.68) and PSYRATS Delusions (M=9.17, SD= 11.43; 95% CI, 12.76 to 16.91) and at week 6 (end of intervention) (PANSS Positive Scale (M=12.50, SD=7.06; 95% CI, 8.01 to 16.99), PANSS Negative Scale (M=11.67, SD= 7.97; 95% CI, 6.60 to 16.73), PANSS General Psychopathology (M=22.75, SD=12.85; 95% CI, 14.59 to 30.91), PSYRATS Voices (M=14.83, SD= 3.27; 95% CI, 1.90 to 16.43) and PSYRATS Delusions (M=3.75, SD= 7.11; 95% CI, -0.77 to 8.27)).

2) Conclusions: the study succeeds in its aim and it is appropriate to now consider seeking funding for a larger study.

Response: We have added to the manuscript “based on this feasibility and acceptability data we plan to submit a funding application, for a larger appropriately powered trial with an internal pilot to investigate the clinical and cost-effectiveness of the TechCare intervention” (Page 17)

3) Limitations: The intervention does not appear to address auditory hallucinations or negative symptoms: this may be perfectly appropriate but needs to be specified and, for the future, inclusion/exclusion criteria adjusted.

Response: This was a preliminary investigation of the TechCare intervention, with the aim of determining feasibility. On the advice of service user researcher (NC) we decided to initially only look at paranoia and low mood symptoms in participants. Based on the findings of this preliminary work we plan to use the full PANSS in the future effectiveness trial, which will include other psychotic symptoms such as hallucination and negative symptoms.

Amendment: We have added to the manuscript "Furthermore, the study only addressed low mood/paranoia in this assessment of feasibility, future research will examine the full spectrum of symptoms of psychosis such as hallucinations and negative symptoms". (Page 16)

4) 'A randomised controlled trial design with a larger sample size, may have strengthened the research design, providing data on the acceptability of the allocation and randomisation procedures' - this is not really a limitation as it wouldn't have strengthened the design at this stage of development.

Response: We agree with the reviewer that a larger sample size may not have been a limitation at this stage, and we have now amended the section to reflect this.

Amendment: We have added to the manuscript "One of the limitations of the study was a small sample size n=16." (Page 16)